# Correlation between Fatty Acid Profile and Anti-Inflammatory Activity in Common Australian Seafood by-Products

**DOI:** 10.3390/md17030155

**Published:** 2019-03-06

**Authors:** Tarek B. Ahmad, David Rudd, Michael Kotiw, Lei Liu, Kirsten Benkendorff

**Affiliations:** 1Marine Ecology Research Centre, Southern Cross University, Lismore 2480, Australia; Tarek.Ahmad@usq.edu.au (T.B.A.); david.rudd@monash.edu (D.R.); 2Division of Research & Innovation, University of Southern Queensland, Toowoomba 4350, Australia; Michael.Kotiw@usq.edu.au; 3Monash Institute of Pharmaceutical Sciences, Monash University, Parkville 3052, Australia; 4Southern Cross Plant Science, Southern Cross University, Lismore 2480, Australia; Ben.liu@scu.edu.au

**Keywords:** seafood waste, polyunsaturated fatty acid, NO inhibition, fish oil, marine nutraceuticals

## Abstract

Marine organisms are a rich source of biologically active lipids with anti-inflammatory activities. These lipids may be enriched in visceral organs that are waste products from common seafood. Gas chromatography-mass spectrometry and fatty acid methyl ester (FAME) analyses were performed to compare the fatty acid compositions of lipid extracts from some common seafood organisms, including octopus (*Octopus tetricus*), squid (*Sepioteuthis australis*), Australian sardine (*Sardinops sagax*), salmon (*Salmo salar*) and school prawns (*Penaeus plebejus*). The lipid extracts were tested for anti-inflammatory activity by assessing their inhibition of nitric oxide (NO) and tumor necrosis factor alpha (TNFα) production in lipopolysaccharide (LPS)-stimulated RAW 264.7 mouse cells. The lipid extract from both the flesh and waste tissue all contained high amounts of polyunsaturated fatty acids (PUFAs) and significantly inhibited NO and TNFα production. Lipid extracts from the cephalopod mollusks *S. australis* and *O. tetricus* demonstrated the highest total PUFA content, the highest level of omega 3 (ω-3) PUFAs, and the highest anti-inflammatory activity. However, multivariate analysis indicates the complex mixture of saturated, monounsaturated, and polyunsaturated fatty acids may all influence the anti-inflammatory activity of marine lipid extracts. This study confirms that discarded parts of commonly consumed seafood species provide promising sources for the development of new potential anti-inflammatory nutraceuticals.

## 1. Introduction

Acute and chronic inflammation is the basis of many serious diseases including asthma, cardiovascular diseases, and rheumatoid arthritis [1]. The stimulation of macrophages during the inflammatory response gives rise to overproduction of several pro-inflammatory mediators, including nitric oxide (NO) via inducible nitric oxide synthase (iNOS) [2]. NO overproduction can lead to tissue damage through cytokine-mediated processes. This molecule can also cause vasodilation, edema, and cytotoxicity [3,4]. Macrophage stimulation also leads to the overproduction of many cytokines including TNFα and interleukins (IL). These pro-inflammatory cytokines have many roles, including the recruitment and activation of more macrophages, effects on the endothelial cells in the blood vessels, as well as playing a role in the perception of pain generated from inflammation [5]. Cytokines such as TNFα are powerful pro-inflammatory mediators during infection, trauma, or surgery, which can trigger short- and long-term effects on the peripheral and central nervous system, leading to exacerbated pain processed by directly affecting specific receptors on sensorial neurons [6,7]. Thus, these pro-inflammatory cytokines and NO are reliable markers for screening new anti-inflammatory treatments *in vitro* and *in vivo*.

The conventional management of inflammation relies mainly on the use of steroidal and non-steroidal anti-inflammatory drugs (NSAIDs). Both drug families are well-known for their common and serious side effects [8], especially when associated with long term consumption, such as is often required for the treatment of chronic inflammatory diseases. Consequently, there is a critical need to identify new sources of less harmful treatments, particularly for the management of diseases associated with chronic inflammation. Our increased understanding of the impact of food and diet on health has driven the search for novel natural medicines [9]. A recent study has shown that patients suffering from chronic inflammatory diseases are more likely to seek out natural anti-inflammatory agents with the intention to minimize the side effects associated with long term use of steroid and NSAIDs [8]. Therefore, the development of new safer anti-inflammatory nutraceuticals is of clinical interest and could have a significant impact on the treatment of inflammatory cases. Functional foods and marine extracts provide a relatively untapped source of potential anti-inflammatory agents, but claims need to be evidence-based.

In comparison to saturated fats, dietary polyunsaturated fatty acids (PUFAs) can have a number of positive impacts on health when incorporated into the diet to meet deficiencies from sub-optimal dietary intake. The human body relies on food as a source of long chain PUFAs, as it is unable to synthesize PUFAs larger than 18 carbons [10]. Both omega 3 (ω-3) alpha-linolenic and omega 6 (ω-6) linoleic PUFAs are considered essential for mammals and can only be obtained from the diet. Seafood (fish and shellfish) lipids are the main sources of biologically active ω-3 long chain PUFAs [10,11]. These PUFAs are known to minimize the occurrence and severity of chronic inflammatory conditions [12,13,14,15], cancer [16,17,18], obesity [19,20], and cardiovascular diseases [10,21,22]. Dietary PUFAs have been shown to improve the quality of life for people suffering from chronic inflammatory diseases such as arthritis, asthma, and neuroinflammatory diseases [11]. Long chain ω-3 PUFAs can directly inhibit inflammation by competing with arachidonic acid or indirectly by affecting the transcription factors or nuclear receptors responsible for inflammatory gene expression [23].

Docosahexaenoic acid (C22:6 ω-3) (DHA) and eicosapentaenoic acid (C20:5 ω-3) (EPA) are the most valuable long chain PUFAs and are considered potent anti-inflammatory agents as a result of the amount and type of the eicosanoids they generate, which interfere with intracellular signaling pathways, transcription factors, and gene expression mechanisms [24]. DHA has been shown to reduce the levels of IL-1β and TNFα in LPS-stimulated peripheral blood mononuclear cells (PBMNC) *in vitro* [25]. Furthermore, studies have shown that a number of health problems including increased inflammatory processes [26], poor fetal development, and a higher risk of Alzheimer’s diseases are associated with diets low in EPA and DHA PUFAs [27].

Omega 3 PUFAs can be commercially sourced from oily fishes including salmon, sardine, and mackerel [10]. There is some evidence that seafoods from temperate Australian origins contain higher amounts of DHA compared to seafood from the northern hemisphere [28]. Some of the PUFA-rich Australian marine organisms include crustaceans, such as school prawn and tiger prawn, oily fishes such as sardine and salmon, and mollusks, including octopus, squid, shelled gastropods, and bivalves [28]. In Western countries like Australia, there is a significant level of waste from underutilization of parts of seafood, as only the choice flesh is consumed by many people. However, valuable PUFAs might not only occur in the predominantly eaten components of the seafood organisms, but could also be present in the undervalued presumptive waste tissues of fish and shellfish. For example, in fish, lipids are not only stored in the subcutaneous tissue, belly flap, and muscle, but are also high in mesenteric tissue, head, and liver [29]. A number of previous studies have investigated the quality of fatty acids in seafood waste tissues. For example, high levels of ω-3 rich PUFAs were found in lipid extracts from the head (26%), intestine (24%), and liver (23%) from the sardine *Sardinella lemuru* [30], and similarly in the tuna *Euthynnus affinis* (head 28.77%; intestine 27.43%; liver 23.98%) [31]. Significant amounts of the valuable ω-3 EPA and DHA were also found in eye orbital samples of tuna in an Australia-based study [28,32]. Therefore, the byproducts of the seafood industry could be a source of high-quality anti-inflammatory fatty acids.

To date, investigations on the anti-inflammatory activity of fish oil have been insufficiently investigated, and consequently, results remain inconclusive [33]. Nevertheless, there is some clinical evidence to support the benefits of krill oil in the treatment of inflammatory conditions [34,35], suggesting that the specific PUFA composition may be important for anti-inflammatory activity. Many molluscan products and derivatives are used in traditional medicines for the treatment of inflammatory conditions [36], and mollusks are also known to be rich in beneficial PUFAs [37]. Examples of lipid extracts from mollusk with demonstrated *in vivo* and *in vitro* anti-inflammatory activity include the New Zealand green-lipped mussel *Perna canaliculus* [38], *Filopaludina bengalensis* foot [39], Indian green mussel *Perna viridis* [40,41], *Mytilus unguiculatus* (Hard-shelled mussel) [40], and sea hares *Aplysia fasciata* and *Aplysia punctata* [42]. Despite the promising anti-inflammatory activity of lipid extracts from mollusks, the only natural anti-inflammatory nutraceuticals available over-the-counter as anti-inflammatory medications are Lyprinol and Biolane sourced from the lipid extract of the New Zealand green-lipped mussel *Perna canaliculus* [38], and Cadalmin^TM^, the lipid extract from a closely related species of bivalve *Perna viridis* in India [41,43]. The anti-inflammatory activities of lipid extracts from predatory cephalopod mollusks are yet to be tested.

This study aims to investigate the composition of lipid extracts from common Australian seafoods including oily fish, prawns, and cephalopods, with a comparison of the edible flesh and under-utilized by-products. The anti-inflammatory activity of these lipid extracts was compared using *in vitro* assays for NO and TNFα inhibition and the inhibition concentrations (IC_50_s) were correlated to the fatty acids composition to provide further insight into how the fatty acid compositions of marine lipid extracts influence the anti-inflammatory potential.

## 2. Results

### 2.1. Comparison of Fatty Acid Composition from Lipid Extracts of Different Seafood and Waste Products

As expected for oily fish, the highest yield of lipid extract was recovered from the Australian sardine, followed by salmon (>100 mg/g tissue, Figure 1A). Substantially lower quantities were recovered from the cephalopods and prawns (5–40 mg/100 g tissue, Figure 1A). Higher quantities of oil were recovered from the viscera and/or heads of all species in comparison to the flesh. The lipid extracts from the cephalopod mollusks had the highest proportion of PUFAs comprising over 40% of the fatty acid composition (Figure 1B, Table 1). Salmon had the lowest percentage of PUFAs (<25%), but the highest percentage of MUFAs, with over 45% of the total fatty acids (Figure 1B and Table 1). Permutational analysis of variance (PERMANOVA) revealed a significant difference between species in the composition of fatty acid classes (Pseudo *F* = 16.4, *p* = 0.001). Pair-wise analysis revealed that the octopus, squid, prawns, and sardines were not significantly different in the relative proportion of fatty acid classes (*p* > 0.05); however, salmon was significantly different to octopus (*p* = 0.0091), squid (*p* = 0.0066) and sardines (*p* = 0.0017). Specifically, there were significantly less saturated fatty acids and more monounsaturated fatty acids in salmon compared to squid (SFA *p* = 0.0035, MUFA, *p* = 0.0054) and sardines (SFA *p* = 0.001, MUFA = 0.0031). Salmon had significantly less PUFAs than octopus (*p* = 0.0144), squid (*p* = 0.0144), and sardines (*p* = 0.0202). The PUFAs were also significantly lower in prawns compared to octopus (*p* = 0.0302) and squid (*p* = 0.0276). The waste products (viscera and heads), did not have a significantly different profile to the more frequently consumed cephalopod flesh or fish fillets, but the prawn heads and viscera had less MUFA, relative to SFA and PUFA, compared to the body flesh (Figure 1B). All extracts had a healthy ω-6/ ω-3 ratio of less than or close to 1 (Table 1B), with the ratio as low as 0.1 in sardines and the flesh of octopus. The ratio of saturated to unsaturated fatty acids was also less than 1 for all species.

Multivariate comparison of the overall fatty acid profiles in the oil extracts revealed significant differences between the species (Pseudo-F = 9.59 *p* = 0.0007), but not between the different types of tissue (Pseudo-F = 2.73, *p* > 0.05), and there was no significant interaction between these factors (Pseudo-F = 1.6, *p* > 0.05). Pair-wise tests confirmed that *S. salar* (salmon) has a different fatty acid composition to all species except prawns (*p* < 0.01), which is driven by a higher percentage of oleic, linoleic, and eicosatrienoic acids in the salmon and prawn heads (Figure 2 and Table 1A). The squid contained higher proportions of stearic acid, arachidonic (ARA), and docosahexaenoic acid (DHA), and were significantly different to sardines (*p* = 0.015), which, along with octopus and prawn bodies, have a higher percentage of the SFA arachidic and the ω-3 PUFA, EPA (Figure 2).

The amounts of EPA, DPA, and DHA per 100g of the seafood tissue were estimated from the yield of oil in the original tissue (Table 1B). Due to high oil yields, the sardines were the best source of these ω-3 PUFAs, with a total amount of over 3500mg/100g tissue in the flesh and over 6000 mg/100 g in the viscera. The viscera of octopus and heads of salmon also had high ω-3 yields with totals of over 1000mg/100g tissue. In all species, the viscera and/or head waste streams produced larger amounts of EPA, DPA, and DHA (Table 1B).

### 2.2. Cytotoxicity

At 50 µg/mL, none of the seafood extracts caused any reduction in cell viability for either 3T3-ccl-92 fibroblasts or RAW 264.7 macrophages (Table 2).

### 2.3. NO Inhibition

Lipid extracts from all Australian seafood organisms demonstrated significant NO inhibition in LPS-stimulated RAW 264.7 cells compared to the solvent positive control, except the lipid extract from school prawn bodies, which only showed weak NO inhibition (Appendix A and Table 2). Lipid extracts from the octopus showed strong NO inhibition with the lowest IC_50_s of 65 µg/mL. The lipid extracts from the waste by-products showed higher NO inhibition (lower IC_50_s) than the more commonly consumed flesh for octopus, squid, salmon, and prawns. All of the Australian seafood lipid extracts were active at much lower concentrations than the reference nutraceutical oils, Lyprinol and Deep Sea Krill oil, at the maximum concentration (Table 2). In fact, Lyprinol showed no inhibition of NO in this assay at the maximum solubility.

Multivariate RELATE analysis using a Spearman rank correlation demonstrated a significant relationship between NO inhibition and fatty acid composition (Rho = 0.302, *p* = 0.0477, 9999 permutations). BEST analysis revealed that a combination of four fatty acids explained the greatest amount of variation in NO inhibiting activity, with a correlation coefficient of 0.428. NO inhibition correlated with lower levels of oleic acid (C18:1) and higher levels of C18:3n-6, C20:2 and C22:2. Univariate correlations to investigate the relationship between IC_50_ for NO and the amount of particular fatty acids in the extracts revealed different trends depending on the fatty acid class (Appendix A). Negative relationships (i.e., lower IC_50_ at higher concentrations = stronger activity) were observed for total SFAs (R^2^ = 0.4) and PUFA (R^2^ = 0.3), as well as ω-3 PUFAs (R^2^ = 0.3), EPA (R^2^ = 0.4), and to a lesser extent, DHA (R^2^ = 0.2). Conversely, MUFAs showed a positive relationship (higher IC_50_s = weaker activity) (R^2^ = 0.3), along with ω-9 FAs (R^2^ = 0.3). Surprisingly NO activity increased with higher SFA:UFA ratios (R^2^ = 0.4), driven largely by inactive MUFAs, but decreased with higher ω-6:ω-3 ratios (R^2^ = 0.3), as expected. There was no relationship between NO inhibitory activity and the amount of ω-6 PUFAs, the ω-3 DPA, or unidentified components in the extracts (Appendix A).

### 2.4. TNF-Alpha Inhibition

Lipid extracts from all the seafood samples demonstrated significant TNFα down-regulatory effects reducing the levels of TNFα in the RAW 264.7 supernatant (Appendix A, Table 2). Octopus viscera again showed the lowest IC_50_ (51 µg/mL), whereas the body of prawns had the highest IC_50_ at ~200 µg/mL. The extract from seafood by-products showed greater TNFα inhibition than the edible flesh for all species (Table 2), and this was most noticeable in prawns, with an IC_50_ nearly three times lower in the heads and viscera that are routinely discarded in Australia. The fish viscera and heads were approximately twice as active as the fillets.

There was a significant correlation between NO inhibition and TNFα inhibition (R^2^ = 0.631), although the Australian sardine fillet had lower TNFα activity than would have been predicted from the NO inhibition. Multivariate RELATE analysis revealed a marginally significant relationship between TNFα inhibition and fatty acid composition (Rho = 0.269, *p* = 0.058, 9999 permutations). BEST analysis identified only two fatty acids, with a correlation coefficient of 0.334. TNFα inhibition was weakly correlated with lower levels of linolenic acid (C18:2) and higher levels of stearic acid (C18:0). Univariate correlations investigating the relationship between IC_50_ for TNFα and the amount of particular fatty acids in the extracts (Appendix A) revealed similar trends to NO inhibition (Appendix A). Decreasing TNFα IC_50_ with higher quantities were observed for total SFAs (R^2^ = 0.3) and PUFA (R^2^ = 0.4), as well as ω-3 PUFAs (R^2^ = 0.4), DHA (R^2^ = 0.3), and to a lesser extent, EPA (R^2^ = 0.2). MUFAs again showed the reverse trend, along with ω-9 FAs (R^2^ = 0.3). TNFα IC_50_ decreased with higher SFA:UFA ratios (R^2^ = 0.2), but increased with higher ω-6:ω-3 ratios (R^2^ = 0.2). There was again no relationship between TNFα IC_50_ inhibitory activity and the amount of ω-6 PUFAs, the ω-3 DPA, or unidentified components in the extracts (Appendix A).

## 3. Discussion

This study demonstrates the quality and anti-inflammatory activity of lipids extracted from different Australian seafood. Seafood is known to be high in PUFAs, which have been previously associated with anti-inflammatory activity. All of the extracts tested in this study contain a high content of PUFAs, with ω-6/ω-3 ratios less than one. Simopoulos [44] found that lower ratios are desirable for reducing the risk of many chronic diseases, with ratios < 4:1 reducing mortality from chronic disease, and ratios less than 3:1 suppressing inflammation due to arthritis. We found that lower ω-6/ω-3 ratios correlated with higher NO and TNFα inhibitory activity across a range of seafood extracts. Furthermore, Western diets typically contain excessive levels of saturated fats and omega 6 fatty acids, which promote the pathogenesis of many diseases, including inflammatory conditions [44]. Our lipid extracts from Australia seafood all had saturated to unsaturated fatty acid ratios of less than 1, but higher amounts of MUFAs rather than SFAs were related to lower NO and TNFα inhibitory activity. Overall, the entire fatty acid composition appears to influence anti-inflammatory activity *in vitro*. Nevertheless, all of our extracts provided a good source of ω-3 fatty acids and significantly inhibited LPS stimulated NO and TNFα production by macrophages *in vitro*. This indicates the potential to value-add the Australian seafood industry based on high-quality marine oils with anti-inflammatory activities.

Anti-inflammatory fatty acids were not only found in the flesh that is normally consumed in Australian seafood, but are also present in high quantities in unprocessed parts like the head and viscera. In fact, the viscera and heads produced a higher yield of oils and contained higher quantities of commercially important long-chain ω-3 PUFAs EPA, DPA, and DHA with known healthy attributes for seafood consumers. The yields of these ω-3 PUFAs in the under-utilized/non-processed parts were substantially higher than in the edible flesh for most species (e.g., ten times the DHA in octopus viscera compared to flesh; four times the amount of EPA in prawn heads compared to flesh; and nearly double the EPA, DPA, and DHA in salmon heads and sardine viscera/heads compared to the flesh). This data is consistent with previous studies which have demonstrated high-quality fatty acid profiles in the uneaten tissues of mackerel tuna fish *Euthynnus affinis*, ray-finned fish *Sardinella lemuruand,* and Alaska pink salmon *Oncorhynchus gorbuscha* [30,31,45]. Overall the yields of EPA and DHA are similar to the range previously reported for mollusks, fish, and crustaceans (e.g., Reference [46]), although the Australian sardine is particularly notable for containing over 1000mg/100g tissue of both EPA and DHA in both the flesh and viscera. Australian seafood waste streams could therefore be used to generate a sustainable source of high-value marine lipids if they can be rapidly processed in a centralized facility to prevent oxidation and degradation.

All the lipid extracts from Australian seafood tested in this study showed significant inhibition of NO and TNFα, except those from the body of school prawns. As a neurotransmitter, NO is a potent inflammatory mediator, as well as playing a role in wound healing and maintaining blood pressure [47]. However, there are many diseases associated with the overproduction of NO, including liver cirrhosis, rheumatoid arthritis, infection, autoimmune diseases, and diabetes [48]. Similarly, TNFα is an important pro-inflammatory mediator that can lead to damaging effects, including neuropathic pain, when over-expressed [49]. Inflammation and neuropathic pain are complex problems involving many mediators and coupled signaling pathways which reduce the effectiveness of single compounds for drug development [49]. Natural extracts that contain a mixture of potential inhibitors of inducible NO Synthase (iNOS), TNFα expression, and other inflammatory pathway modulators might be effective for controlling chronic inflammation. For example, studies on the NZ Green-lipped mussel extract Lyprinol^®^, in a rat model for arthritis, have demonstrated that it modulates inflammatory cytokines (TNFα, IL-6, IL-1α, and IL-γ) and decreases the synthesis of some proteins associated with inflammation, whilst increasing malate dehydrogenase synthesis, which is related to glucose metabolism [50]. The effects on regulatory proteins were proposed to reduce energy for MHC-1 activation as a novel mechanism of action with anti-inflammatory efficacy at lower doses than other fish oil preparations. Lyprinol is a patented combination of 50 mg of PCSO-524^®^ (lipid extract from *P. canaliculus*), 100 mg of a proprietary oleic acid blend, and 0.225 mg of vitamin E, so the ω-3 fatty acids in this mussel lipid extract may act synergistically with the anti-oxidant Vitamin E. Similarly, krill oil contains the antioxidant astaxanthin which can prevent lipid peroxidation, thus preserving of the ω-3 fatty acids EPA and DHA, in addition to acting directly on a number of biomarkers [51]. Krill oil has been shown to modulate cytokines, lipidogenesis, lipid peroxide, oxidative enzymes, glucose metabolism, and the endocannabinoid system in a range of animal studies [52]. It is possible that some of the unidentified components in our extracts have anti-oxidant activity and/or immunomodulatory activity that complements or enhances the activity of ω-3 fatty acids. Further *in vivo* studies investigating a range of anti-inflammatory markers and modulators will be required to establish the mechanism of action and novel potential of these Australian seafood lipid extracts.

The anti-inflammatory effects of fish oils and krill oil are typically attributed to long chain ω-3 PUFAs [14,52]. Similarly, we found a correlation between the amount of ω-3 PUFAs in the extracts and the IC_50_s for both LPS stimulated NO and TNFα inhibition in RAW264.7 macrophages. In particular, the concentrations of both EPA and DHA were found to correlate with higher activity, but with EPA showing a stronger relationship with NO inhibition and DPA explaining more of the variation in TNFα inhibition. Previous studies on pure DHA and EPA have confirmed that these ω-3 PUFAs strongly inhibit NO production (IC_50_ < 25 μM) [53] and inducible NO synthase in LPS stimulated RAW264 macrophages (DHA at 30μM and EPA at 60 μM) [54]. Both EPA and DHA (100 μM) have also been shown to reduce TNFα secretion in LPS stimulated RAW264.7 cells after 24 hr exposure to 100 μM [55], and they significantly inhibit the secretion and transcription of TNFα in LPS stimulated THP-1 macrophages at 25 mM [56]. These studies lend support to the idea that EPA and DHA are contributing to the *in vitro* anti-inflammatory activity in our extracts, however, without further purification and testing against the pure compounds, we cannot conclude that there are no other factors involved. Indeed, higher concentrations of saturated fatty acids were also found to correlate with NO and TNFα inhibition, and our multivariate correlations indicate that the overall composition of fatty acids is important, along with potentially unidentified factors.

In this study, lipid extracts from the two cephalopod mollusks contained the highest proportion of ω-3 PUFAs and were associated with relatively high NO and TNFα inhibition. This supports the use of the flesh from octopus (*Zhang Yu*) and squid (*Qiang Wu Zei*) in traditional Chinese Medicine for inflammatory conditions [36]. Our comprehensive review of the anti-inflammatory, wound healing, and immunomodulatory activity of mollusks [36] found no previous *in vitro* studies on cephalopod extracts, and only a handful of *in vivo* animal models that have tested the ink and melanoprotein of squid [57,58]. Nevertheless, one previous study found that cuttlefish (*Ommastrephes bartrranii*) liver oil significantly reduced formalin and carrageenan-induced paw edema in rats fed 1% cuttlefish liver oil for 45 days [59]. We found that the cephalopods produce a fairly low yield of oil in comparison to oily fish, and consequently, they had lower overall amounts of EPA, DPA, and DHA per mg of tissue. This suggests the cephalopods may not be as good a functional food for dietary intake of ω-3 PUFAs based on the mass obtainable from the wet weight of edible tissue as compared to some other seafood, such as the Australian sardines, which have very high yields of ω-3 PUFAs in their flesh. However, the viscera and heads of octopuses and squids produce a significant waste stream that could be value-added based on their high-quality anti-inflammatory oils. For example, a large proportion of Southern jig squid fishery is sold as processed tubes in Australia, generating approximately 48% viscera as waste from ~1000 t annual harvest [60]. This justifies further investigation into cephalopod lipid extracts for anti-inflammatory applications.

The fatty acid profiles of the cephalopod and sardine lipid extracts are dominated by ω-3 PUFAs, whereas the salmon and prawns contain relatively high MUFA and ω-9 PUFAs. The high NO activity of sardine and cephalopod extracts reflects their richness in ω-3 fatty acids, especially DPA, DHA, and EPA, which are arguably the healthiest PUFAs [12,23,61]. The main ingredients of the well-known anti-inflammatory nutraceutical Lyprinol^®^ and Biolane^®^, lipid extracts from the New Zealand green-lipped mussel (GMLE) *Perna canaliculus*, are long chain ω-3 PUFA. The percent of the ω-3 PUFA in GMLE is about 37.1% of the total fatty acids [40]. A similar proportion of ω-3 PUFAs was found in the cephalopod mollusk extracts in this study. Interestingly, however, the different levels of ω-3 fatty acids in the various seafood extracts were not found to correlate directly with either NO or TNFα inhibitory concentrations in this study. Rather, lower levels of the MUFA oleic acid and higher levels of the ω-6 acids gamma-linolenic (C18:3), eicosadienoic (C20:2) and docosadienoic (C22:2) correlated with higher NO activity, whereas lower levels of linolenic (C18:2) and higher levels of the SFA stearic acid (C18:0) explained more of the variation in TNFα inhibitory concentrations. This implies that the overall composition of fatty acids in seafood oils could influence the anti-inflammatory activity, with particular fatty acids having either beneficial or antagonistic effects. This may help explain the variable outcomes from clinical trials on the use of fish oils for some inflammatory conditions [62,63].

The minimal anti-inflammatory activity detected in our prawn flesh extract was similar to that observed for a commercial krill oil in the same assays. The fatty acid composition of our prawn extracts was similar to that previously reported for a commercial krill oil (10% ω-3 PUFAs, 14% ω-6 PUFAs and 35% MUFAs), which was found to significantly modulate inflammation and lipid metabolism in mice transgenic for TNFα [33]. Antarctic krill oil has also been shown to inhibit LPS-induced iNOS in a rodent model [64] and to protect against rheumatoid arthritis in mouse models, but without effects on serum cytokines [35]. Further research on Penaeid extracts is therefore justified despite the relatively low *in vitro* activity. In particular, the waste streams from aquacultured prawns can be sustainably produced in comparison to wild harvested krill, although their fatty acid compositions could be impacted by an artificial diet [65].

Lyprinol^®^, the anti-inflammatory nutraceutical composed of lipid extracts from the Green-lipped mussel *Perna canaliculus,* was also used as a reference drug but did not show detectable inhibition of NO and TNFα in our *in vitro* assays. Green-lipped mussel extracts have been previously found to suppress iNOS expression and inhibit NO production in LPS-induced RAW264.7 cells by regulating nuclear factor kappa B [66], as well as inhibiting TNFα in LPS-stimulated human THP-1 monocytes [67]. Therefore, the lack of activity in both the commercial marine oils we tested is likely due to their relative insolubility in ethanol as a result of product formulation. Furthermore, *in vitro* assays do not always provide a good predictor of *in vivo* activity. Nevertheless, given the wealth of evidence relating to the beneficial effects of ω-3 PUFAs in clinical trials, the preliminary *in vitro* anti-inflammatory activity observed here, along with the beneficial fatty acid composition of Australian seafood extracts, justifies further research to value-add the industry by developing a sustainable supply of high-quality fish oil.

## 4. Materials and Methods

### 4.1. Chemicals and Reagents

*Escherichia coli* LPS (O128:B12, Sigma-Aldrich, Castle Hill, Australia), sulfanilic acid, N-(1-Naphthyl) ethylenediamine (NED), 85% orthophosphoric acid sodium nitrite (NaNO_2_), and HPLC grade solvents were obtained from Sigma Aldrich (St. Louis, MO, USA). The mouse TNFα ELISA kit was purchased from BD biosciences (Sparks, MD, USA). Penicillin–streptomycin solution, Dulbecco’s Modified Eagle’s Medium (DMEM), fetal bovine serum (FBS), sodium pyruvate, and l-glutamine were from Life Technology Australia (Mulgrave, VIC, Australia). The two cell lines, RAW264.7 mouse macrophages and 3T3 Swiss albino (ATCC^®^ CCL92™) cell lines, were purchased from the American Type Culture Collection (ATCC^®^, Manassas, VA, USA).

### 4.2. Sample Collection

Fresh seafood used in this study including octopus (*Octopus tetricus* n = 3), Australian sardine (*Sardinops sagax* n = 4), salmon (*Salmo salar* n = 3), school prawn (*Penaeus plebejus*, n = 12), and squid (*Sepioteuthis australis* n = 3) were purchased fresh from Ballina seafood co-op., Ballina, Australia. Salmon and squid viscera are processed during harvesting to prevent degradation and fouling of the prime edible flesh; however, the heads of these organisms are still available as a waste stream. All other species were obtained whole and the typically consumed flesh was separated from the waste streams, including internal organs (viscera) and heads. Lyprinol (BLACKMORES^®^, Alexandria, Australia) and Krill oil (Swisse, Collingwood Melbourne, Australia) were purchased from a local pharmacy.

### 4.3. Lipid Extraction

Extracts from the cephalopods, *O. tetricus* and *S. australis*, included the tentacles and mantle tissue (edible flesh) and body viscera, comprised of the gastrointestinal tract and other internal organs for *O. tetricus* and just the heads for *S. australis*. Extracts from the fish included flesh fillets and viscera, including heads from *S. sagax*, and just the head from *S. salar*. The head with viscera (waste) and body (edible flesh) were also extracted from the school prawn *Penaeus plebejus*. Lipids were extracted as above using a solvent to tissue ratio of 19 mL final volume for every 1 g tissue.

The solvent homogenates were vacuum filtered through Whatman paper (No. 1) into separating funnels. Saturated NaCl solution (6.2 M) was added to the solvent phase to a final ratio of 8:4:3 (Chloroform: methanol: NaCl solution). The organic phases were collected and the solvent evaporated on a rotary evaporator (Rotavapor^®^ R-114; BÜCHI Labortechnik AG, Flawil, Switzerland). Extracted lipids were transferred into glass vials, dried under a stream of nitrogen gas, weighed on an analytical balance to calculate yield per g tissue, and then covered and stored in minimal hexane at −80 °C until required.

### 4.4. Fatty Acid Methyl Ester (FAME) Analysis

Subsamples of 200 μL of foot and viscera lipid extracts from all species were placed in 10 mL pyrex glass vials for derivitization. NaCl in methanol solution (0.5 M, 1.5 mL) was added under nitrogen gas, capped and shaken for 10 secs. Samples were then heated at 100°C for 10 min in a dry block and cooled. Two mL of 14% Boron trifluoride in methanol was added and bubbled with nitrogen gas for 8 s, then placed in a 100 °C dry block for 30 min. After cooling, 1 mL of hexane containing 1 μg 2,6-Di-tert-butyl-4-methylphenol (BHT) was added, and samples were shaken for 30 s. Saturated NaCl solution (5 mL) was added and the samples were shaken to create an upper lipid layer, which was collected and stored at −80 °C for FAMEs and GC-MS analysis.

FAMEs were analyzed using a GC (Agilent 6890N, Santa Clara, CA, USA) coupled to an Agilent 6890 flame ionization detector (FID) using a BPX 70 capillary column (70% cyanopropyl polysilphenylene-siloxane, 50 m length, 0.22 mm internal diameter and 0.25 µm thickness). The FID was operated at 260 °C and the split injector was maintained at 230 °C. High-purity helium was used as the carrier gas and maintained with a linear flux of 1 mL/min. The GC oven was held at 100 °C for 5 min and then raised to 240 °C at a rate of 5 °C/min. 1 μL of each subsample extract was injected with a split ratio of 200:1 and a column flow of 1 mL/min.

FAMEs were identified by peak retention time and elution order and compared against a reference FAMEs standard test mix (SUPELCO 37-Component FAME Mix CRM47885, Bellefonte, PA, USA) and a marine test mix PUFA No.1 (Marine Source, Analytical Standards, Sigma-Aldrich, Castle Hill, Australia). Some samples were further analyzed using an Agilent gas chromatography-mass spectrometer (GC-MS) with an Agilent 5973 Mass Selective Detector to confirm the identity of the fatty acids. The mass spectra were recorded at 70 eV ionization voltage over the mass range of 35–550 amu. To facilitate the identification of DPA, which was not in the test mix, a soft ionization MS technique at 40 eV ionization voltage was employed to ionize the lipid molecules in the *D. orbita* samples without causing extensive fragmentation. The spectrum was compared on MS databases (WILEY 275 online and NIST98, Gaithersburg, MD, USA), along with retention times and elution order from extensive literature searches including the American Oil Chemists’ Society. The relative composition of each identified fatty acid was calculated by peak integration from the GC [68]. The concentration of each fatty acid was estimated using BHT as an internal standard. In each sample, the area under the curve for each fatty acid was calibrated against the peak area for BHT and adjusted for molecular weight, then scaled for the concentration on 1mg per g extract. For the ω-3 PUFAs EPA, DHA, and DPA, we also calculated the yield per 100 g of tissue by adjusting for the amount of crude lipid extract obtained per g of tissue that was extracted.

### 4.5. Cell Lines and Cell Culture

The Murine RAW264.7 macrophages and 3T3 fibroblast cell lines were obtained from American Type Cell Culture (ATCC). Both cell lines were maintained in 10% FBS supplemented DMEM, 100 µg/L streptomycin, and 100 IU/mL penicillin at 37 °C and 5% CO_2_ atmosphere. Cells were passaged every 48–72 h [69].

### 4.6. Lipid Extract Preparation

Lipid extracts were dried under nitrogen gas flow before being weighed and dissolved in HPLC-grade 100% ethanol. The stock solutions of the lipid extracts were diluted in color-free DMEM before being added to the cell culture, and the final concentration of ethanol in all experiments was around 0.35%. Stock solutions of the lipid extracts were prepared fresh on the day of the experiment prior to addition to the cell culture. The solubility of all extracts in the cell cultures was confirmed under an inverted microscope (200 and 400×). Each sample was tested in triplicate and each experiment was repeated independently at least three times on different days.

### 4.7. Cytotoxicity Assay

Toxicity of the lipid extracts used in this study was assessed using a crystal violet cytotoxicity assay as previously described by Feoktistova, et al. [70] using both the RAW 264.7 macrophages and 3T3 ccl-92 fibroblasts cell lines. Briefly, cells were seeded at a density of 2 × 10^4^ cells/well in a 96-well plate and then incubated for 18–24 h. Lipid extracts were added then and incubated for 24 h before the media was aspirated and the cells washed twice in a gentle stream of water. Water was removed by tapping the plate on a pile of paper towel followed by addition of 50 µL of 0.5% crystal violet staining solution and incubated for 20 min at room temperature. The plate was then washed 4 times with water and air dried for 2 h. Finally, 200 μL of methanol was added to each well and incubated for 20 min at room temperature on a rocker. The optical density was then measured at 570 nm using Anthos Zenyth 200rt plate reader (Anthos Labtec Instruments, Heerhugowaard, Netherlands). Chlorambucil in a gradient concentration was used as a positive control.

### 4.8. NO Inhibition Assay

The production of NO by LPS stimulated RAW 264.7 macrophages was measured in the cell culture supernatant using the Greiss reaction method as previously described by Ahmad et al. [64]. In brief, RAW 264.7 macrophages were seeded at a density of 10^6^ cell/mL and incubated overnight. The following day, cells were incubated with different concentrations of the extracts 50, 25, 12.5, 6.25, or 3.125 µg/mL 1 h prior to LPS stimulation. Twenty-four h after LPS stimulation, the supernatant was collected, and an equal volume of supernatant and Greiss’ reagent was mixed in a 96 well plate and incubated in dark for 10–15 min. Absorbance was read at 550 nm using an Anthos Zenyth 200rt plate reader (Anthos Labtec Instruments, Heerhugowaard, Netherlands). Sodium nitrite was used as a standard in this assay, and all assays were repeated in triplicate. The commercially available marine nutraceuticals Lyprinol (BLACKMORES^®^, Sydney, Australia) and Deep Sea Krill oil (Swisse, Collingwood, Australia) were used as reference anti-inflammatory nutraceuticals at the same test concentrations. Dexamethasone at 2.5 µM concentration was used as a reference drug. All assays were repeated three times.

### 4.9. TNF Alpha Inhibition Assay

The levels of TNFα produced by LPS stimulated RAW 264.7 macrophages in the cell culture supernatant were measured using a mouse TNFα ELISA kit (R&D Systems, Minneapolis, MN, USA) and performed as per the manufacturer’s instructions. All extracts were tested in five different concentrations 50 µg/mL, 25 µg/mL, 12.5 µg/mL, 6.25 µg/mL, and 3.125 µg/mL. Dexamethasone was used as a reference anti-inflammatory drug, and untreated, stimulated cells (LPS + ethanol) were used as a positive control. Absorbance was read at 450 nm using an Anthos Zenyth 200rt plate reader (Anthos Labtec Instruments, Heerhugowaard, Netherlands). The commercially available marine nutraceuticals Lyprinol^®^ (BLACKMORES^®^, Alexandria, Australia) and Deep Sea Krill oil (Swisse, Collingwood Melbourne, Australia) were used as reference anti-inflammatory nutraceuticals. The assays were repeated in triplicate.

### 4.10. Statistical Analysis

PRIMER v 7 + PERMANOVA software (version 7, Primer-e, Albany, New Zealand) was used to explore the multivariate differences in lipid profiles, and univariate analyses were used to test differences in specific lipid classes and anti-inflammatory activity between the various seafood extracts. Separate Euclidean distance similarity matrices were created for the fatty acid percent composition, the composition of fatty acid classes, and the totals for each fatty acid class (SFA, MUFA, PUFA and ω-3, ω-6, n-9, ω-3:ω-6 and DMAAs), as well as the IC_50_ for NO inhibition and TNFα inhibition. Principle Coordinate Ordination plots were generated on the fatty acid composition with vector overlay based on Pearson’s correlation with a cut-off at r > 0.8 to identify which fatty acids contributed most to the separation between samples.

To assess the relationship between fatty acid composition and NO inhibition or TNFα inhibition, relate analyses were undertaken on PRIMER V7 using Spearman rank correlation and 9999 permutations. This was followed by a BIOENV stepwise model on the Euclidean distance similarity matrix to identify which set of fatty acids explained the most variability for each of the anti-inflammatory markers.

## 5. Conclusions

In conclusion, lipid extracts from Australian marine seafood were found to contain a high ratio of unsaturated: saturated fatty acids and significant anti-inflammatory activity. The inhibition of NO and TNFα in LPS stimulated macrophages was correlated with higher levels of SFAs and PUFA, and in particular the ω-3 PUFAs EPA and DHA, as well as with lower levels of MUFAs, thus indicating that the overall composition of marine lipid extracts can influence the anti-inflammatory activity. High valued marine oils rich in healthy ω-3 PUFAs were not only demonstrated in the edible parts of seafood, but the under-utilized components of these organisms showed similar if not higher proportions of PUFAs and anti-inflammatory activity. In particular, the byproducts from cephalopod mollusks appear to have good anti-inflammatory activity, with potential for the development of another high-quality marine oil for nutraceutical applications. Further *in vitro* and *in vivo* studies are required to optimize and develop the seafood waste stream as used for natural anti-inflammatory treatments.

## Figures and Tables

**Figure 1 marinedrugs-17-00155-f001:**
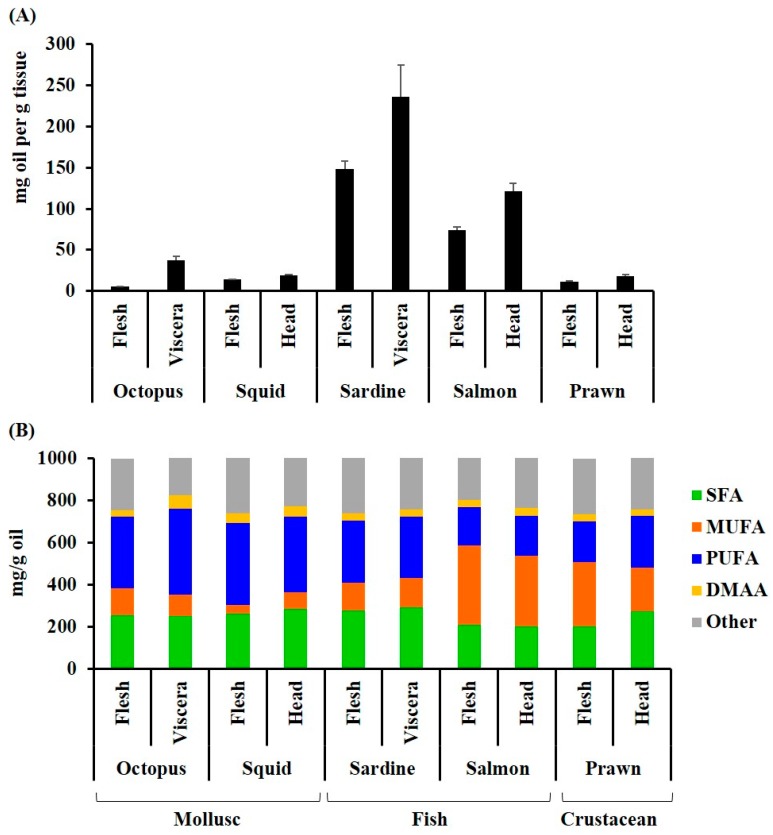
Lipid composition of some common Australian seafood flesh and waste streams: (**A**) The amount of oil extracted from the flesh (mg/g tissue); (**B**) the amount of the main fatty acid classes and other hydrocarbons (dimethyl acetal aldehydes) in the lipid extracts (mg/g oil). The fatty acids were quantified by GC-FID and identified against reference standards with supplementary GC-MS analyses. The samples are: *Octopus tetricus* flesh and viscera, *Sepioteuthis australis* (squid) flesh and head; *Sardinops sagax* (Australian sardine) flesh and viscera, including heads; *Salmo salar* (Atlantic salmon) flesh and head (SH); and *Penaeus plebejus* (Australian school prawn) flesh and head, including viscera.

**Figure 2 marinedrugs-17-00155-f002:**
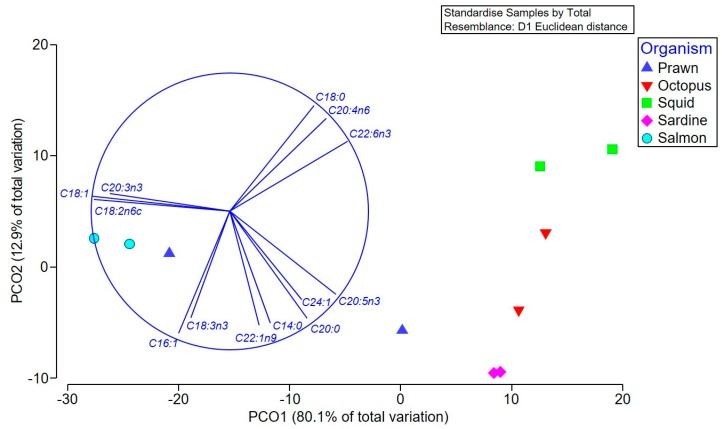
Principal coordinate ordination (PCO) of the fatty acid composition of various Australian seafood species. Vector overlay based on the Pearson correlation (r > 0.8) identifies the main fatty acids contributing to the separation between extracts, with higher levels of the specifically labeled fatty acids occurring in samples in the direction of the vector.

**Table 1 marinedrugs-17-00155-t001:** Fatty acid profiles of lipid extracts from the commonly consumed flesh (tentacles, fillet, body) and waste products (viscera, heads) of Australian seafood organisms: (**A**) μg fatty acid per mg oil extract (estimated from a 2,6-Di-tert-butyl-4-methylpheno (BHT) internal standard and adjusted for molecular mass); (**B**) percent composition of dimethyl acetal aldehydes and major fatty acid classes in the lipid extract, as well as the estimated quantity of eicosapentanoic (EPA) and docosahexanoic (DHA) per 100g tissue for each seafood.

**(A) Fatty Acid**	**Trivial Name**	***Octopus tetricus***		***Sepioteuthis australis***		***Sardinops sagax***		***Salmo salar***		***Penaeus plebejus***	
		**Flesh**	**Viscera**	**Flesh**	**Head**	**Flesh**	**Viscera & Head**	**Flesh**	**Head**	**Flesh**	**Head & Viscera**
**Saturated Fatty Acids (SFAs)**											
C12:0	Lauric	0.7	0	0	0	0.8	0.8	0.8	0.8	0.7	0
C13:0	tridecanoic	0	0	0	0	0.7	0.8	0	0	0	1.6
C14:0	myristic	43.0	20.0	11.3	14.8	57.8	59.7	16.5	20.1	17.8	16.1
C15:0	pentadecanoic	4.1	4.6	4.2	5.9	5.0	6.0	1.6	2.3	5.7	41.0
C16:0	palmitic	146.9	132.0	160.9	176.4	163.1	173.0	142.9	132.2	121.8	112.3
C17:0	heptadecanoic	8.0	14.0	11.1	11.6	65.6	6.9	3.1	4.0	7.4	26.0
C18:0	Stearic	43.7	69.1	69.4	67.2	34.6	37.0	40.1	39.4	39.2	44.5
C20:0	arachidic	3.6	4.1	1.5	1.5	4.5	4.8	0.8	0.8	1.5	3.2
C21:0	henicosanoic	0.7	0	0.8	0	0.8	0.8	0.8	0.8	3.1	21.4
C22:0	Behenic	2.2	2.5	1.6	2.4	2.4	1.7	0.9	0.8	1.6	3.4
C23:0	tricosanoic	0	0	0	0	0	0	1.0	0.9	0.9	1.9
C24:0	lignoceric	6.2	2.6	1.6	4.2	0.8	0.9	0.9	0.9	0.8	1.7
**Monounsaturated Fatty Acids (MUFAs)**											
C14:1	myristoleic	6.2	2.3	0	0	1.5	1.5	1.6	1.5	0.7	2.3
C15:1	pentadecanoic	0.7	1.5	0	0	0.7	0.8	0.8	0.7	0.7	1.5
C16:1	palmitoleic	44.1	22.5	5.6	10.9	59.4	60.7	48.5	42.3	39.8	36.8
C17:1	heptadecanoic	0.7	1.5	0.7	2.2	0.7	1.5	2.3	2.2	7.0	50.6
C18:1n9t	elaidic	2.1	3.1	1.4	3.0	0.7	0.8	0.8	0.7	1.5	3.9
C18:1n9c	oleic	42.2	43.7	16.7	55.9	50.3	54.2	301.9	261.7	230.3	93.4
C20:1n9	eicosenoic	19.2	14.4	12.6	0	3.8	3.9	15.6	16.8	16.3	8.0
C22:1n9	erucic	8.2	7.6	1.6	4.0	9.5	9.1	4.3	4.8	5.5	5.0
C24:1n9	nervonic	5.3	6.0	1.6	2.5	4.8	5.0	1.7	2.4	3.2	4.3
**Polyunsaturated Fatty Acids (PUFAs)**											
C18:2n6c	Linoleic (LA)	12.3	8.5	2.8	0	15.2	0.8	82.8	76.3	67.2	20.8
C18:3n6	Ƴ-linolenic (GLA)	1.4	5.4	0.7	11.2	2.2	2.3	1.6	2.2	2.2	1.5
C18:3n3	α-linolenic (ALA)	10.6	4.7	0.7	2.2	13.9	14.5	10.3	8.9	8.6	3.1
C20:2	eicosadienoic	1.4	5.5	2.2	12.2	1.5	1.6	4.8	7.5	6.6	6.3
C20:3n3	eicosatrienoic	2.2	3.3	2.3	4.8	3.1	3.3	4.2	4.7	5.4	5.0
C20:4n6	arachidonic (ARA)	21.6	56.7	77.1	63.2	11.4	12.0	4.8	7.0	10.5	34.0
C20:5n3	eicosapentanoic (EPA)	120.3	107.0	62.5	53.6	123.2	125.6	14.0	19.6	26.2	75.3
C22:2	docosadienoic	1.5	0.9	1.6	0	0.8	0.8	0	0.8	0.8	1.7
C22:5n3	docosapentanoic (DPA)	24.4	22.9	10.1	7.9	18.1	18.0	7.5	12.2	14.4	33.8
C22:6n3	docosahexanoic (DHA)	141.9	192.9	229.8	205.7	105.1	111.0	48.0	50.7	51.9	66.7
**(B) Fatty Acid**	**Trivial Name**	***Octopus tetricus***		***Sepioteuthis australis***		***Sardinops sagax***		***Salmo salar***		***Penaeus plebejus***	
		**Flesh**	**Viscera**	**Flesh**	**Head**	**Flesh**	**Viscera & Head**	**Flesh**	**Head**	**Flesh**	**Head & Viscera**
**Dimethyl Acetal Aldehydes**											
dimethyl acetal octadecan-1-al		0	2.5	3.0	2.9	0	0	0	0	0	0
dimethyl acetal nonadecan-1-al		3.3	4	1.4	1.8	3.6	3.7	3.4	3.6	3.9	3.0
**Categories**											
SFAs		25.5	24.9	26.2	28.4	27.7	29.2	20.9	20.3	20.0	27.3
MUFAs		12.8	10.3	4.0	7.8	13.1	13.8	37.8	33.3	30.5	20.6
PUFAs		33.7	40.8	39.0	36.1	29.4	29.0	17.9	19.0	19.3	24.8
Total ω-3		29.9	33.1	30.5	27.4	26.3	27.2	8.4	9.6	10.7	18.4
Total ω-6		3.5	17.1	8.6	7.4	2.9	1.5	9.1	8.5	8.0	5.6
Total ω-9		15.0	7.5	3.4	6.5	6.9	7.3	32.4	28.6	25.7	11.5
Total unidentified		24.2	17.6	26.4	23.0	26.1	24.3	20.0	23.8	26.2	24.3
Saturated/unsaturated ratio		0.6	0.5	0.6	0.6	0.7	0.7	0.4	0.4	0.4	0.6
ω-6/ω-3 ratio		0.1	0.2	0.3	0.3	0.1	0.1	1.1	1.0	0.7	0.3
EPA per 100 g tissue		64.2	395.7	88.7	99.3	1845.5	2972.5	104.2	238.5	29.9	138.2
DPA per 100 g tissue		13.1	84.7	14.3	14.6	268.8	426.0	55.8	148.5	16.4	62.0
DHA per 100 g tissue		75.7	713.4	326.0	381.2	1560.7	2627.0	357.2	617.0	59.1	122.4

**Table 2 marinedrugs-17-00155-t002:** Cytotoxicity and anti-inflammatory activity of the lipid extracts from various tissues of commercial seafood species, as well as two commercially available marine oils, calculated from the average of three repeat assays.

Organism	Extract	3T3 ccl-92 Fibroblasts Viability at 50 µg/mL	RAW 264.7 Macrophages Viability at 50 µg/mL	NO Inhibition IC_50_ (µg/mL)	TNFα Inhibition IC_50_ (µg/mL)
***Octopus tetricus*** (Octopus)	Viscera	100%	100%	64.6	51.0
	Flesh	100%	100%	71.2	71.0
***Sepioteuthis australis*** (Squid)	Head	100%	100%	91.1	67.7
	Flesh	100%	100%	114.2	78.8
***Sardinops sagax*** (Australian Sardine)	Viscera/head	100%	100%	84.6	71.1
	Fillet	100%	100%	66.5	147.7
***Salmo salar*** (Salmon)	Head	100%	100%	97.3	85.8
	Fillet	100%	100%	157.9	157.1
***Penaeus plebejus*** (School Prawn)	Head/viscera	100%	100%	88.0	71.2
	Body	100%	100%	306.4	201.7
***Euphausia superba***	Krill Oil	100%	100%	337.8	99.8
***Perna canaliculus*** (NZ Green-Lipped Mussel)	Oil (Lyprinol)	100%	100%	No detectable activity	>> max test dose 587.9

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
