# Peer review of "Correlation between Fatty Acid Profile and Anti-Inflammatory Activity in Common Australian Seafood by-Products"

_marinedrugs, 2019, doi:10.3390/md17030155_

Round 1

Reviewer 1 Report

There are too many scientific short cuts : line 61 and 67 : not all PUFA have positive impacts on human health line 199 : to be good for health ratio w6/w3 muste be far less than one, provide the ratios you get in comparison with the WHO recommendations. Line 226 : either you write might be instead of may be, either you provide scientific proofs of your point of view (clinical studies on humas for instance) line 227 an: what is the composition of lyprinol ? Omega3 have been demonstrated to be anti-inflammatory, not w6. So is there a need for more precisions.  Same for line 232 and followings. You need to be more accurate line 83 : provide ratio w3/w6 and w3/totat lipids or w3/saturated I do not understand figure 2. So can it be improved with table1 and provide for instances the amount of w3 PUFA, w6, mono-instaurated, saturated  for 100gr of tissue with a "classical" control. Why were the viscera not analyzed for salmon, and the head not done for sardino and Penaeus ?  It is well known that lipids are very present in ipdophil oranges like liver and brain for instance. For NO and TNF : the resuslts can not be interpretated with a proper inflammation on endotoxin LPS, palmitate (TLR4 ligand) of your extracts. You need to proof that their quality is high enough + check TLR/NOD? NLR expression on your cell lines. How did you check the lipids effectively reached the cells and did not stay as droplet ? You need to be more accurate in displayiong you results (provide values) : LPS, nothing, vehicle, lipid extract. -line 391 provide information on their composition and whether they are clean or not (for an immunologist, i.e as previously mentioned toxinq, LPS, palmitate ...) In the conclusion section : what is your conclusion ? that current wastes may be of interest for w3 purification, and provide a more sustainable approach?

Author Response

We appreciate the opportunity to revise our manuscript in response to the reviewers comments. The feedback has been very useful for improving the rigour and presentation of our data.  Our responses to specific comments are detailed below in red text under each comment, from each reviewer in turn. We have also tracked changes on the revised manuscript. Line numbers refer to the tracked version of the manuscript unless otherwise stated.  I hope you find these responses satisfactory and the revised manuscript acceptable for publication in Marine Drugs

Response to reviewer 1:

There are too many scientific short cuts

line 61 and 67 : not all PUFA have positive impacts on human health

Author response: The details on specific PUFAs that are beneficial for human health are provided in lines 65-81. However, we have revised the introductory statement in this paragraph to explain more precisely that

“In comparison to saturated fats, dietary polyunsaturated fatty acids (PUFAs) can have a number of positive impacts on health, when incorporated into the diet to meet deficiencies from sub-optimal intake”. (lines 62-64).

An increased ratio of PUFA:SFA in the diet is beneficial to human health irrespective of whether all individual PUFAs have positive impacts. Furthermore, the health benefits include obtaining essential fatty acids that humans can’t synthesize and these do include some omega 6 acids that have antagonistic or no anti-inflammatory properties, as well as omega 3 acids that are known for anti-inflammatory and other bioactive properties. This is explained in the text in the paragraph that follows the introductory sentence.

line 199: to be good for health ratio w6/w3 muste be far less than one, provide the ratios you get in comparison with the WHO recommendations

Author response: Line 199 discusses the outcomes of a statistical analysis comparing TNF inhibition to fatty acid composition. I think the reviewer is probably referring to original Line 208.

The WHO does not provide any recommendation for ratios of w-6/w-3 PUFAs - in fact the only recommendations from WHO that we could find are in regards to regular consumption of fish with serving equivalent of 200-500mg of EPA and DHA. The new results included in Table 1B demonstrate that the cephalopods and fish tested in our study can provide these amounts of EPA and DHA. The cephalopods and sardines do also have w-6/w-3 ratios much less than one. However, according to the National Institute of Health factsheets on omega 3 fatty acids “the optimal ratio - if any- has not been defined” for w-6:w-3  (https://ods.od.nih.gov/factsheets/Omega3FattyAcids-HealthProfessional/#en10)

Simopoulos (2002) explains that the optimal ratio depends on the specific disease or complication, but recommends w-6:w-3 ratios of less than 4:1 due to a 70% decrease in total mortality from cardiovascular disease and ratio of 2-3:1 effectively suppressed inflammation in patients with rheumatoid arthritis. Therefore the ratio of w-3:w-6 does not need to be far less than one to be good for human health and is more often well above 1. However, to clarify the statement in the opening paragraph of our paper we have revised the text as follows:

“All of the extracts tested in this study contain a high content of PUFAs, with ω-6/ω-3 ratios less than one... Simopoulos [44] found that lower ratios are desirable for reducing the risk of many chronic diseases, with ratios < 4:1 reducing mortality from chronic disease and ratios less than 3:1 suppressing inflammation due to arthritis. We found that lower ω-6/ω-3 ratios correlated with higher NO and TNFα inhibitory activity across a range of seafood extracts. Furthermore,…… all had a saturated to unsaturated fatty acid ratio of less than 1, but higher amounts of MUFAs, rather than SFAs were related with lower NO an TNFα inhibitory activity. Overall the entire composition appears to influence anti-inflammatory activity in vitro. Nevertheless, all of our extracts provide a good source of ω-3 fatty acids and significantly inhibited LPS stimulated NO and TNFα production by macrophages in vitro.” (Lines 245-255).

Line 226: either you write might be instead of may be, either you provide scientific proofs of your point of view (clinical studies on humas for instance)

Author response: We have revised the text to “might be” instead of “may be” and we have also deleted “more” - so that it now just suggests that a mixture of potential inflammatory inhibitors and modulators “might be effective for controlling chronic inflammation…” . (Line 228). In regards to the “scientific proof” - it needs to be appreciated that this sentence is worded cautiously, such that it is posed as an hypothesis or suggestion rather than a statement of fact, and ultimately there is no real proof in science - just support for or against specific hypotheses. Some supporting evidence for this suggestion is provided in the following sentences where we provide examples from the green-lipped mussel in a rat model and krill oil from a review of in vivo animal studies. There are complex human ethics issues that complicate similar clinical studies on humans. Nevertheless, clinical studies in humans are not actually required to support the idea that extracts that target multiple inflammatory markers can be effective for controlling inflammation. We also clearly acknowledge the need for further in vivo studies on our extracts in the last sentence in this paragraph. (lines 301-302)

line 227 an: what is the composition of lyprinol? Omega3 have been demonstrated to be anti-inflammatory, not w6. So is there a need for more precisions. Same for line 232 and followings.

Author response: In the paragraph that the reviewer is referring to here, we were discussing mode of action and cellular multiple targets in marine anti-inflammatory extracts rather than fatty acid compositions. To clarify this, we have changed “PUFA preparations” to “fish oil preparations” in line 245. We have also added information on Lyprinol (lines269-272):

“Lyprinol is a patented combination of 50 mg of PCSO-524® (lipid extract from P.canaliculus), 100 mg of a proprietary oleic acid blend and 0.225 mg of vitamin E, so the ω-3 fatty acids in muscle lipid extract may act synergistically with the anti-oxidant Vitamin E..” (Line 292-295)

And krill oil (lines 297-299) “Similarly, krill oil contains the antioxidant astaxanthin which can prevent lipid peroxidation, thus preserving of the ω-3 fatty acids EPA and DHA, in addition to acting directly on a number of biomarkers in animal studies [50].”

You need to be more accurate line 83: provide ratio w3/w6 and w3/totat lipids or w3/saturated

I do not understand figure 2.

Author response: Figure 2 is a standard multivariate PCO plot showing the differences between the different seafood species based on the overall fatty acid composition. As explained in the Figure legend the vectors show which fatty acids are contributing to the difference between species. So for example, there are higher levels of C18:1, C203 and C18:2 in salmon, and higher levels of C18:0, C20:4 and C22:6 in squid. This is explained in the text referring to the figure line141-146. We have slightly revised the Figure 2 legend to explain:

“Vector overlay based on Pearson correlation (r > 0.8) identifies the main fatty acids contributing to separation between extracts, with higher levels of the specifically labelled fatty acids occurring in samples in the direction of the vector.” (Lines 170-172)

We have now also added univariate correlations, which more readers will be familiar with in Supplementary Figures S2 and 4S. These results are now explained in Lines 208 -217 and Lines 233-240.

So can it be improved with table1 and provide for instances the amount of w3 PUFA, w6, mono-instaurated, saturated for 100gr of tissue with a "classical" control.

Author response: The full details of the fatty acid composition are presented in Table 1, including totals for SFA, MUFA, PUFA, w-3, w-6 and w-9 acids, as well as the w-6:w-3 ratios. We had initially based this data on percent composition of the fatty acids and other compounds detected in the GC-FID. However, we have now converted all of this data to mg/g lipid extract, estimated using BHT as an internal standard to calibrate the areas under the curve, then adjusted for the molecular weight of the fatty acid -without the methyl groups. This was the only way we could quantify the fatty acids because there was no other internal standard of known concentration added to extracts that were analysed. It would not be appropriate to prepare new extracts for analysis because the fatty acids composition is correlated to the anti-inflammatory activity. Hence it is essential that the fatty acids are quantified from exactly same extracts that were also used in the bioassays.

Table 1A now provides the estimated amount of each fatty acid as mg/g lipid, whereas Table 1B summarises the main classes of fatty acid and other components in the extracts as a percent of the total extract. The trends in this data are very similar to the original Table (based on percent composition from the area of peaks in the GC-FID), with the exception that the amount of “other” or unidentified components is larger. This is because the lipid extracts include some compounds that are not detected in the GC-FID. The unidentified components are now calculated by subtracting the sum of all identified compounds from the total lipid extract (1mg). We believe this provides a more accurate indication of the proportion of major fatty acid classes in the whole lipid extract and thank the reviewer for this suggestion.

It should be noted that we have elected not to present all this data as mg/100g tissue because we are more interested in comparing the fatty acid composition of the lipid extracts to the anti-inflammatory activity in the extracts, to assess the relative quality of the different seafood oils. The amount in the tissue is more relevant to seafood quality for consumption, whereas it could still be worth extracting good quality oils from low yielding species with large waste streams, even if the overall yield is lower than some other species with lower quality oils. Nevertheless, we do agree the amount of oils in the tissue is important, so we have now added the yield of lipid extract (oil in mg per gram tissue) to Figure 1 part A (page 4). Part B of Figure 1 has also been updated to show the composition of fatty acid classes and other components as mg/g oil. Figure 1 legend has been updated (line 148-149)

“Figure 1: Lipid composition of some common Australian seafood flesh and waste streams: A) amount of oil extracted from the flesh (mg/100g tissue); B) Amount of the main fatty acid classes and other hydrocarbons (dimethyl acetal aldehydes) in the lipid extracts (mg/g oil)….”

New text summarising the yields of lipid extract have been added to the results (line 123-126):

“As expected for oily fish, the highest yield of lipid extract was recovered from the Australian sardine, followed by salmon (> 100mg/100g tissue, Figure 1A). Substantially lower quantities were recovered from the cephalopods and prawns (5-40mg/100g tissue, Figure 1A). Higher quantities of oil were recovered from the viscera and/or heads of all species in comparison to the flesh.”

We have also added new data to Table 1B on the yields of EPA, DHA and DPA, in mg per 100g tissue. These results are described in lines 175-180:

“The amount of EPA, DPA and DHA per 100g of the seafood tissue was estimated from the yield of oil in the original tissue (Table 1B). Due to high oil yields, the sardines were the best source of these ω-3 PUFAs, with a total amount of over 3500mg/100g tissue in the flesh and over 6000mg/100g in the viscera. The viscera of octopus and heads of salmon also had high ω-3 yields with totals of over 1000mg/100g tissue. In all species, the viscera and/or head waste streams produced larger amounts of EPA, DPA and DHA (Table 1B).”

We have discussed these new results in lines 262-275

“In fact, the viscera and heads produced a higher yield of oils and contained higher quantities of commercially important long chain ω-3 PUFAs EPA, DPA and DHA with known healthy attributes for seafood consumers. The yield of these ω-3 PUFAs in the under-utilised/non-processed parts was substantially higher than in the edible flesh for most species  (e.g. 10 x the DHA in octopus viscera compared to flesh; four times the amount of EPA in prawn heads compared to flesh; and nearly double the EPA, DPA and DHA in salmon heads and sardine viscera/heads compared to flesh)….. Overall the yield of EPA and DHA are similar to the range previously reported for molluscs, fish and crustaceans [e.g. 46], although the Australian sardine is particularly notable for containing over 1000mg/100g tissue of both EPA and DHA in both the flesh and viscera).”

In lines 299-301 we acknowledge the potential activity from unidentified compounds in the extracts: “It is possible that some of the unidentified components in our extracts have anti-oxidant activity and/or immunomodulatory activity the complements or enhances the activity of ω-3 fatty acids.

And in Lines 328-336

“We found that the cephalopods produces a fairly low yield of oil, and consequently lower overall amounts of EPA, DPA and DHA per mg of tissue compared to oily fish like the sardines. This suggests the cephalopods may not be as good functional foods for dietary intake of ω-3 PUFAs, based on the mass obtainable from the wet weight of edible tissue, as compared to some other seafood, such as the Australian sardines, which have very high yields of ω-3 PUFAs in their flesh. However, the viscera and heads of octopus and squid produce a significant waste stream that could be value-added based on their high quality oils. For example, a large proportion of squid fishery is sold as processed tubes in Australia; generating approximately 48% viscera as waste from 1000 t annual harvest” (Australian Bureau Agricultural and Resource Economics an Sciences http://www.agriculture.gov.au/abares).

Why were the viscera not analyzed for salmon, and the head not done for sardino and Penaeus ? It is well known that lipids are very present in ipdophil oranges like liver and brain for instance

Author response: We used only available waste streams for testing: in Australia, some parts of fish (for some species) are processed "on-farm" or "at-sea", to prevent degradation and pungent odours transferring to the edible flesh. This means some parts of the waste are not readily available for further processing. In particular, salmon and squid viscera is processed during harvesting to prevent fouling, but heads are available as a waste stream and include the brain and some other internal organs. The octopus, prawns and Australian sardines were obtained whole. Our prawn heads did include viscera and conversely, the sardine viscera included the head. For the octopus we dissected out the internal tissue and separated this from all the external edible flesh (including tentacles). We had just labelled each sample according to the dominant component and apologise for this confusion. The manuscript has now been revised to correct and clarify all these inconsistencies as follows:-

Figure 1 and associated Figure legend (Lines 152-155)

Table 1 column headings page 6 & 7 - added head or viscera where required

Table 2 page 6 added head or viscera to the extract column where required.

Results text: Line 140“prawn heads and viscera”.  Line 225 “heads and viscera”

Methods: Line 391-395 “Salmon and squid viscera is processed during harvesting to prevent degradation and fouling of the prime edible flesh, however the heads of these organisms are still available as a waste stream. All other species were obtained whole and the typically consumed flesh was separated from the waste streams including internal organs (viscera) and heads.” 

Lines 400-404 “Extracts from the cephalopods, O. tetricus and S. australis, included the tentacles and mantle tissue (edible flesh) and body viscera, comprised of gastrointestinal tract and other internal organs for O. tetricus and just the heads for S. australis. Extracts from the fish included flesh fillets and viscera, including heads from S. sagax, and just the head from S. salar. The head with viscera (waste) and body (edible flesh) were also extracted from the school prawn Penaeus plebejus.

For NO and TNF : the resuslts can not be interpretated with a proper inflammation on endotoxin LPS, palmitate (TLR4 ligand) of your extracts. You need to proof that their quality is high enough + check TLR/NOD? NLR expression on your cell lines.

Author response: We are unsure what the reviewer means by this comment. The inhibition of NO and TNFa in LPS stimulated cells are very well established as useful in vitro screens for anti-inflammatory activity in the literature. The aim of our study was to establish whether the fatty acid composition of the seafood extracts correlated with inhibition of these two inflammatory markers. We have not extrapolated our results beyond these in vitro studies. However, we do discuss the broader literature on krill oil and green-lipped mussel extracts with reference to a range of other inflammatory markers. Furthermore, the anti-inflammatory effects of EPA and DHA are already well established in the literature from both in vitro and in vivo studies. This is now detailed in lines 309-314 of the discussion. At this stage we are unable to do further experiments to assess other inflammatory markers. However, we believe that our results on the composition and activity of the seafood waste streams and the novel correlation of fatty acid composition to inhibition of these two inflammatory marker is significant enough to warrant publication in Marine Drugs.

How did you check the lipids effectively reached the cells and did not stay as droplet?

Author response: Initial tests with DMSO and ethanol were used to find the most suitable solvent for extract delivery to the cells. All extracts applied to cells were visually inspected using an inverted microscope (200X and 400X) to confirm the solubility and exposure to the cell lines used in the study. Insoluble extracts in our previous studies have been easily identified by precipitates or oil droplets on the surface. In this study we can confirm that no oil droplets were present from the extracts dissolved in ethanol in any of the test assays, including the maximum concentrations tested. We have added the following sentence to the methods line 457-458:

“The solubility of all extracts in the cell cultures was confirmed under an inverted microscope (200 and 400x)”.

You need to be more accurate in displayiong you results (provide values): LPS, nothing, vehicle, lipid extract.

Author response: We have added two supplementary Figures showing the level of NO and TNF inhibition of all extracts at all test concentrations relative to the LPS stimulated solvent control. Supplementary Figure S1 is cited in line 196 and Supplementary Figure S3 is cited in line 221.

line 391 provide information on their composition and whether they are clean or not (for an immunologist, i.e as previously mentioned toxinq, LPS, palmitate ...)

Author response: The original Line 391 of the methods is referring to the commercial products Lyprinol and Krill oil. These are over the counter nutraceuticals sold by pharmacies in Australia specifically for human use. The manufacturers and suppliers have to be approved by the Therapeutic Goods Administration with full quality control regulations. As we did not manufacture these products, we are unable to provide specific details in their composition and quality control, however we have provided the suppliers for these commercially available products.

In the conclusion section : what is your conclusion ? that current wastes may be of interest for w3 purification, and provide a more sustainable approach ?

Author response: Yes our main conclusion is that  the under-utilised waste streams from these Australian seafood species could be value-added as nutraceuticals that are high in anti-inflammatory w-3 PUFAs. We have not commented on sustainability in comparison to other sources, because that depends on the sustainability of the entire fisheries industries and is outside the scope of this paper. However, these fisheries are closely monitored and managed in Australia.

Another important conclusion from our study is that the entire fatty acid composition and not just w-3 PUFAs can influence the inhibition of NO and TNFα produced by stimulated macrophages. This has important implications for refining seafood oils for anti-inflammatory applications. Therefore, we have added the following sentence to the conclusion (Lines 513-517)

“The inhibition of LPS stimulated NO and TNFα in LPS stimulated macrophages was correlated with higher levels of SFAs and PUFA, and in particular the ω-3 PUFAs EPA and DHA, as well as with lower levels of MUFAs, thus indicating that the overall composition of marine lipid extracts can influence the anti-inflammatory activity. “

Author Response

We appreciate the opportunity to revise our manuscript in response to the reviewers comments. The feedback has been very useful for improving the rigour and support for conclusions drawn in the paper.  Our responses to specific comments are detailed below in red text under the comment. We have also tracked changes on the revised manuscript. Line numbers refer to the tracked version of the manuscript unless otherwise stated.  responses to both reviewers are in the attached file. I hope you find these responses satisfactory and the revised manuscript acceptable for publication in Marine Drugs.

Reviewer 2

In line 108, they stated that” anti-inflammatory nutraceuticals available … Lyprinol…” and included it as a reference for their anti-inflammatory assay. Unfortunately, Lyprinol showed no efficacy as an anti-inflammatory nutraceutical with their assays (Table 2). Therefore, it is clear that the lipid extracts of Australian octopus and squid has strong anti-inflammatory activity, but it is unclear whether it is the high content of the w-3 lipids in those two seafood sources. It would be highly informative to determine if purified, pure w-3 PUFAs (as a positive control) exhibit anti-inflammatory activity. Without this kind of positive control (reference as they said), it will remain to be unclear if the Australian octopus and squid have high anti-inflammatory efficacy because of their high w-3 PUFA content or some other ingredients. Here a more explicit conclusion that the authors found two independent aspects or a cause-and-effect.

Author response: In this study Krill oil and Lyprinol were used to provide a comparison to commercially available formulated marine oil extracts - hence we regarded them as reference extracts rather than positive controls. The lack of activity in these commercial extracts appears to be related to solubility issues in cell culture, as discussed in lines 370-371. We agree that these do not substitute for a positive control and indeed it would have been very informative to use EPA and/or DHA as positive controls, but unfortunately it is not possible to add a control retrospectively, because we wouldn’t be using the same batch of cells in culture for a valid comparison of responses. We will certainly take this advice on board in future studies.

Nevertheless, there is substantial evidence in the published literature that pure DHA and EPA strongly inhibits NO production (IC50<25 μM (Ohata, Fukuda, Takahashi, Sugimura, & Wakabayashi, 1997)  and inducible NO synthase in LPS stimulated RAW264 macrophages (DHA at 30μM and EPA  at 60 μM)  (Komatsu, Ishihara, Murata, Saito, & Shinohara, 2003). Both EPA and DHA (100μM) have also been shown to reduce TNFα secretion in LPS stimulated RAW264.7 cells after 24hr exposure to 100μM (Honda, Lamon-Fava, Matthan, Wu, & Lichtenstein, 2015) and they significantly inhibited secretion and transcription of TNFα in LPS stimulated THP-1 macrophages at 25mM (Mullen, Loscher, & Roche, 2010). We have now cited this literature in the discussion Lines 309-314 and use this to support the idea that w-3 PUFAs in our cephalopod extracts are likely to contribute to the observed anti-inflammatory activity, but also acknowledge the need for further purification and testing.

To further explore the relationship between amount of certain fatty acids (including w-3 EPA, DHA and DPA) and NO and TNFα inhibition across all our extracts, we have undertaken some additional correlation analyses with interesting outcomes. We have added the correlation graphs to Supplementary Figure S2 and S4 and the results are summarised in lines 208-217 and 233-240. We have also discussed these results in lines 253-258

“Our lipid extracts from Australia seafood all had a saturated to unsaturated fatty acid ratio of less than 1, but higher amounts of MUFAs, rather than SFAs were related to lower NO and TNFα activity. Overall the entire fatty acid composition appears to influence anti-inflammatory activity in vitro. Nevertheless, all of our extracts provide a good source of ω-3 fatty acids and significantly inhibited LPS stimulated NO and TNFα production by macrophages in vitro.”

And lines 306-311

“The anti-inflammatory effects of fish oils and krill oil are typically attributed to long chain ω-3 PUFAs [14, 52]. Similarly, we found a correlation between the amount of ω-3 PUFAs in the extracts and the IC50s for both LPS stimulated NO and TNFα inhibition in RAW264.7 macrophages. In particular, the concentrations of both EPA and DHA were found to correlate with higher activity, but with EPA showing a stronger relationship with NO inhibition and DPA explaining more of the variation in TNFα inhibition.”

Despite the supporting evidence from the literature and our correlation analyses, we do agree that it is not possible to conclude that omega 3 fatty acids are responsible for the anti-inflammatory activity in our assays. In fact, it is quite plausible that the activity in seafood extracts like ours is due to a combination of bioactive compounds in the lipid extracts. We have added the following sentence to acknowledge that whilst:

“These studies lend support to the idea that EPA and EPA are contributing to the in vitro anti-inflammatory activity in our extracts, however, without further purification and testing against the pure compounds, we can not conclude that there are no other factors involved. Indeed higher concentrations of saturated fatty acids were also found to correlate with NO and TNFα inhibition and our multivariate correlations indicate that the overall composition of fatty acids is important, along with potentially unidentified factors.” (lines 314-319)

Although we can not establish cause and effect in our extracts at this stage, we believe that this doesn’t diminish the potential for value-adding these Australian seafood waste streams. We provide novel information that they are a good source of w-3 PUFAs and that the crude oil extracts inhibit NO and TNFα in vitro. This provides the stimulus for future research aimed at refining the extracts for optimal activity in future in vitro and in vivo studies.

Honda, K. L., Lamon-Fava, S., Matthan, N. R., Wu, D., & Lichtenstein, A. H. (2015). EPA and DHA Exposure Alters the Inflammatory Response but not the Surface Expression of Toll-like Receptor 4 in Macrophages. Lipids, 50(2), 121-129. doi:10.1007/s11745-014-3971-y

Komatsu, W., Ishihara, K., Murata, M., Saito, H., & Shinohara, K. (2003). Docosahexaenoic acid suppresses nitric oxide production and inducible nitric oxide synthase expression in interferon-γ plus lipopolysaccharide-stimulated murine macrophages by inhibiting the oxidative stress. Free Radical Biology and Medicine, 34(8), 1006-1016. doi:10.1016/S0891-5849(03)00027-3

Mullen, A., Loscher, C. E., & Roche, H. M. (2010). Anti-inflammatory effects of EPA and DHA are dependent upon time and dose-response elements associated with LPS stimulation in THP-1-derived macrophages. The Journal of Nutritional Biochemistry, 21(5), 444-450. doi:10.1016/j.jnutbio.2009.02.008

Ohata, T., Fukuda, K., Takahashi, M., Sugimura, T., & Wakabayashi, K. (1997). Suppression of nitric oxide production in lipopolysaccharide-stimulated macrophage cells by omega 3 polyunsaturated fatty acids. Jpn J Cancer Res, 88(3), 234-237.

Simopoulos, A. P. (2002). The importance if the ratio of omega-6/omega-3 essential fatty acids. . Biomedicine and Pharmacotherapy, 56, 365-379.

Round 2

Reviewer 1 Report

The modifications were performed, so is the final version suiting me.

Reviewer 2 Report

The authors responded to the major comments and is now acceptable. It would have been so much better article if they undertook the suggested experiments, but with the current modification of their conclusion it is acceptable.